# Recent Advances in Biosensor Technologies for Point-of-Care Urinalysis

**DOI:** 10.3390/bios12111020

**Published:** 2022-11-15

**Authors:** Chuljin Hwang, Won-June Lee, Su Dong Kim, Sungjun Park, Joo Hee Kim

**Affiliations:** 1Electrical and Computer Engineering, Ajou University, Suwon 16499, Republic of Korea; 2Department of Chemistry, Purdue University, West Lafayette, IN 47907, USA; 3Graduate School of Clinical Pharmacy and Pharmaceutics, Ajou University, Suwon 16499, Republic of Korea; 4Leading Convergence of Healthcare and Medicine, Institute of Science & Technology (ALCHeMIST), Ajou University, Suwon 16499, Republic of Korea; 5College of Pharmacy, Ajou University, Suwon 16499, Republic of Korea

**Keywords:** urine sensor, point-of-care urinalysis, clinical application for urinalysis, dipstick test, biosensors

## Abstract

Human urine samples are non-invasive, readily available, and contain several components that can provide useful indicators of the health status of patients. Hence, urine is a desirable and important template to aid in the diagnosis of common clinical conditions. Conventional methods such as dipstick tests, urine culture, and urine microscopy are commonly used for urinalysis. Among them, the dipstick test is undoubtedly the most popular owing to its ease of use, low cost, and quick response. Despite these advantages, the dipstick test has limitations in terms of sensitivity, selectivity, reusability, and quantitative evaluation of diseases. Various biosensor technologies give it the potential for being developed into point-of-care (POC) applications by overcoming these limitations of the dipstick test. Here, we present a review of the biosensor technologies available to identify urine-based biomarkers that are typically detected by the dipstick test and discuss the present limitations and challenges that future development for their translation into POC applications for urinalysis.

## 1. Introduction

Human urine samples contain several components that can indicate the health condition of a patient, and therefore aid in the diagnosis of common clinical conditions such as diabetes mellitus (DM), urinary tract infections (UTIs), renal stone disease, kidney disorders, liver problems, obesity, other metabolic disorders, or fetal hypothyroidism [1,2,3,4,5,6,7,8,9,10,11,12,13,14,15,16]. Urine samples also provide evidence of underlying health problems in pre-symptomatic and asymptomatic individuals, which can facilitate early intervention and therapy [17,18]. These benefits play an important role in encouraging individual access to continuous and regular health monitoring.

As a point-of-care testing (POCT) analyte, the urine sample is the ideal liquid excretion compared to other biofluids such as saliva, plasma, blood, and serum because it can be collected in large quantities, enabling easy to handle, low risk of infection. Despite these potential benefits of urine samples, to avoid test errors, the following should be thoroughly considered when applied: (i) effects by exogenous substances of urinary metabolites such as water, drugs, and food, (ii) proper storage time and operation during the collection process, and (iii) the complexity, diversity, and strong interference from other compounds in the urine. POCT devices refer to the analysis of a sample near the patient outside of the laboratory and provide test results. Nowadays, POCT devices provide fast and efficient diagnosis, ease of handling, and portability, making them an attractive solution to better healthcare services for patients. In addition, POCT devices can help improve disease diagnosis and management by integrating them into existing healthcare systems. Commonly available POCT urinalysis devices include dipsticks, lateral flow assays, paper-based devices (μPADs), and microfluidic assays.

Among POCT urinalysis devices, a dipstick test is commonly conducted (Figure 1) [19], which provides a method for the verification of several components, including glucose, hydrogen ion, protein, ketone bodies, blood cells, nitrites, bilirubin, leukocyte esterase (LE), urobilinogen, and specific gravity [20,21]. These test strips have small pads infused with different reagents, and rapid color changes reflect the interactions between the reagents and components in the urine samples. Although dipstick tests are simple, quick, and inexpensive, it has certain limitations in terms of sensitivity, selectivity, and reusability.

With respect to sensitivity, the dipstick test—as a point-of-care (POC) diagnostic—is only intended for qualitative detection in situations where accurate measurements of analytes at extremely low concentrations are not as critical. For instance, the dipstick test for proteinuria and hematuria is useful in screening for chronic kidney disease (CKD); however, quantitative methods based on measurement of the urine creatinine concentration are considered to be more effective to assess the severity of proteinuria [22,23,24]. In addition, because the dipstick test is entirely dependent on colorimetry, reflected visually by a color change on the reagent pads, only semi-quantitative analysis is available, where the results provide only an estimate of the amount of substance present [25]. Moreover, the test results manually read by the naked eye are subjective, and differences in lighting conditions or individual color perception can influence the accuracy of the test.

With respect to selectivity, urine samples are complex and contain many components that could trigger false-negative and/or false-positive results in the dipstick test [26]. For example, proteinuria and vitamin C in the urine may produce a false-negative result for the LE test [27]. A false-positive result may be produced for hematuria due to contamination of the dipstick pad with oxidizing agents in urine, such as myoglobin, povidone-iodine, and bacteria [28]. Therefore, it is generally recommended that microscopic analyses be performed to confirm a negative or positive result of the dipstick test [29]. In addition, inorganic ions in urine, (i.e., sodium, potassium, calcium, chloride, and iodine), which play an essential role in the composition of human body fluids, cannot be detected with the dipstick test because reagent reactions with such analytes are not in the visible-light range. The reusability of a POC application is an important property to reduce the cost per test, as well as to facilitate continuous calibration for higher accuracy.

To overcome these limitations of the dipstick test, various biosensor technologies are being developed, showcasing potential in POC applications. Furthermore, advancements in the area of nanoscience and sensing technologies have led to considerable growth in the development of biosensors with desired and highly sought properties such as excellent sensitivity, higher selectivity, and reusability, as compared to the dipstick test (Figure 1). These advantages can help both patients and researchers with a more accurate, efficient, and reliable resolution of health problems.

Here, we present a review of the biosensor technologies available for the identification of urinary biomarkers that are traditionally detected by the dipstick test. This review particularly focuses on the recent advances in biosensor technologies to assess the performance and potential of biosensors as POC applications. A total of 635 studies were obtained through a search of the five databases based such as PubMed, Scopus, IEEE Xplore, Google Scholar, and Embase electronic on our optimized search query in Appendix A (Appendix A), and their filtered publication records from a total of 212 articles for publication trends of biosensors as shown in Appendix A. We then carefully reviewed the full-text versions to check if the articles met the inclusion criteria. Finally, articles with (1) unrelated titles and authors, (2) duplicated titles published in the same year and same journal, (3) an optical sensing system, and (4) review articles were excluded using EndNote, leaving a total of 26 articles for analysis.

We summarize the biosensor features and the factors that affect the performance of urinary biomarkers in terms of the detection method, the limit of detection (LOD), and range of detection compared with those of the dipstick test (Table 1). Finally, we discuss the key challenges and future prospects of biosensors for POC applications.

## 2. Detecting Analytes in Urine for Urinalysis

### 2.1. Detection of Glucose in Urine Sample

According to the International Diabetes Federation, DM is a metabolic disease that affects millions of people worldwide, which is characterized by hyperglycemia with abnormal blood glucose levels [5]. To reduce related complications and manage DM independently, continuous glucose monitoring using various biofluids is needed [56,57]. As opposed to blood glucose tests that require finger pricking to draw blood for each test, which can cause inconvenience and pain to the patient, urine samples can be more conveniently used for glucose measurement. Moreover, the presence of glucose in urine is an indicator of severe DM. Therefore, the use of urine tests for glucose monitoring has attracted substantial attention.

The dipstick test for glycosuria, which is caused by DM, is widely used in clinical practice and public health screening owing to its advantages such as low cost, ease of use, and non-invasiveness; however, its main limitation is low sensitivity [58]. Therefore, for many people, most glycosuria is not diagnosed until severe complications develop, making it more difficult to reduce further deterioration. There has been extensive effort toward developing a biosensor for the detection of glucose in the urine, which would allow patients to track their glucose levels accurately, along with the requirement of frequent glucose monitoring to manage diabetes [59]. For the accurate determination of urine glucose, various detection methods have been employed based on techniques such as fluorescence [60], visible-near infrared spectroscopy [61], surface plasmon resonance [62], and electrochemistry [63]. Among these methods, there has been particularly high interest in the electrochemical sensing technique for urine glucose detection because of its advantages of low cost, fast response time, wide detection range, and ease of use.

A new method was developed to detect glucose based on graphene according to an increasing pH shift due to the enzymatic activity of glucose oxidase (GOx). Fenoy et al. [30] proposed a glucose sensor based on graphene field-effect transistors (GFETs) coupled with enzymes immobilized by electrostatic interaction. GOx has been the most widely utilized enzyme for this purpose owing to its low cost, bioactivity, selectivity, and stability [64]. Among the various proposed strategies for GOx immobilization, the authors used the electropolymerization technique with poly (3-amino-benzylamine-co-aniline) (PABA) to prepare a GFET glucose sensor, as shown in Figure 2a,b. The major advantage of using PABA on GFETs for glucose sensing is the higher quantification ability compared with that possible using conventional glucose sensors that are dependent on electrochemical transduction, allowing for higher reproducibility and the ability to detect glucose in a flow measurement setup. The GOx-PABA-GFET glucose sensor exhibits a wide dynamic range from 10 μM to 1 mM and a low LOD (4.1 μM), enabling the determination of glucose in the urine, as the normal glucose concentration range in urine is between 0 mM (0 mg/dL) and 0.8 mM (15 mg/dL) [65]. In addition, the sensor was able to detect glucose in diluted urine samples with a sensitivity of −13 ± 2 μA per decade of glucose concentration, as shown in Figure 2c,d. Although the GOx-PABA-GFET glucose sensor showed good performance in detecting glucose, it presents limitations with respect to the enzyme’s activity, which can be influenced by factors such as temperature, pH, humidity, and the harsh purification process of the enzyme [66,67].

Huang et al. [31] recently reported glucose biosensors with high sensitivity and selectivity using a polymer and graphene without an enzyme. The biosensor was functionalized with a poly (acrylamide (Aam)-3-acrylamidophenylboronic acid (AAPBA)-N-dimethylaminopropyl acrylamide (DMAAPA)) for glucose, allowing the covalent bonding of AAPBA to its surface via chemisorption. The resulting functionalized surface was found to exhibit the highest sensitivity toward glucose compared with that of the other components in urine, such as carbamide, creatinine, and l-cysteine (Figure 2e). Specifically, the biosensor exhibited a sensitivity of 822 μAcm^−2^mM^−1^, a linear range from 0.04 to 10 mM, and a LOD of 1.9 μM for detecting the presence of glucose in human urine samples. In addition, by the simple process of hydrochloric acid treatment, the biosensor showed excellent reproducibility and repeatability, which was over 93.324% at 0.4 mM of the original value after 20 cycles, as shown in Figure 2f. The authors claim that their platform serves as a polymer-functionalized glucose biosensor, thus providing new opportunities for the development and application of high-performance sensors.

### 2.2. Detection of Hydrogen Ions in Urine Sample

Urine pH, a relatively easily measurable biomarker, is an important indicator of UTI and renal stone disease [6]. In the absence of pathological conditions, the normal range of urine pH is typically between 4.5 and 8, with an average of 6 [68,69]. A urine pH of 8.5 or 9.0 is often suggestive of a UTI-caused urea-splitting bacterium such as Proteus mirabilis or Pseudomonas aeruginosa. At a urinary pH above 6.0, urinary crystallization inhibitors are deficient, leading to calcium phosphate crystals that form insoluble salts as one of the causes of renal stone disease [70], and a urinary pH below 5.5 may induce the formation of anhydrous or dihydrate uric acid crystals when the uric acid concentration is high [71]. Therefore, accurate measurement of urinary pH is essential. Owing to its properties of high accuracy and precision, the gold-standard method for urine pH monitoring is the glass electrode and pH meter [72,73]. However, at present, the urine dipstick is widely used for at-home tests given the various advantages of POC assessments, such as ease of use, convenience, and cost. Nevertheless, there has been substantial debate about the accuracy of such tests [74,75].

Rabboh and O’Neil [32] developed a novel voltametric pH sensor consisting of three-dimensional (3D)-printed graphene/polylactic acid (G/PLA) filament electrodes, which uses intrinsic functional groups such as quinones on the surface of electrodes for pH sensitivity (Figure 3a). Increased quinones that accumulate on the G/PLA electrodes via a multistep pretreatment protocol improved the performance of the sensor, extending the pH measurement range from 2.02 to 11.22. In addition, as the solution pH changes, the anodic and cathodic potential shift was found to be −60 ± 2 mV/pH, which showed good agreement with the theoretical value predicted by the Nernst limit (59.14 mV/pH at 25 °C derived from the Nernst equation), as shown in Figure 3b. Moreover, G/PLA electrodes show excellent agreement for complex samples such as vinegar, cola, urine, serum, and antacid measured with a glass pH probe which is the gold standard for pH measurement (Figure 3c). The major challenge of this biosensor is to achieve the direct sensing of analytes in undiluted complex biological media. To date, only a few biosensor technologies have been developed for the detection of undiluted samples. The authors concluded that their developed electrochemical biosensor has the potential to expand the scope of measuring specified analytes in unadulterated complex samples.

Building upon the invention of the first ion-sensitive field-effect transistor (ISFET) to detect pH values in solution by Piet Bergveld [76], the transistor-based pH sensor was developed, exhibiting promising properties such as high sensitivity, miniaturization, low cost, and long lifetime in various environmental conditions. Many studies have focused on metal oxide transistors for pH measurements such as indium–gallium–zinc-oxide (IGZO) [33], indium oxide (In_2_O_3_) [77], and zinc oxide (ZnO) [78]. Lee and colleagues [33] reported that electrolyte-gated thin-film transistors were very close to the Nernst limit without any additional settings. Figure 3d shows an illustration of the transistor architecture with a conventional top-gate structure on a p-type Si substrate. They used a sol-gel IGZO for the active channel layer, and the source and drain electrodes (Au) were thermally evaporated on the Si substrate to a thickness of 100 nm. As shown in Figure 3e,f, they measured the transfer characteristics of the device with pH values decreasing from 7 to 3 and a sensitivity of 48 mV/pH to investigate the effect of pH on the electrolyte. Although they successfully demonstrated remarkable performance without further chemical functionalization, they noted that more improvements in terms of sensitivity are required for POC application. To enhance the sensitivity, various approaches have been applied to the design of transistor-based pH biosensors. Chae and colleagues [34] proposed the biosensor with amino-silanization on the IGZO surface as a POC application. According to the binding model [79], the positively charged surface of IGZO at an acidic pH leads to the accumulation of negatively charged carriers in the channel, which shifts to a negative threshold-voltage (V_TH_). The sensitivity of the biosensor modified with 3-aminopropyl-triethoxysilane (APTES) was observed to be 65.9 mV/pH from the slope between V_TH_ and pH values in the range of 4 to 10, and this sensitivity was higher than the Nernstian limit with the same pH sensitivity. This study demonstrated that the amine group’s equilibrium with amino-silanized surfaces for changes in an acidic pH promotes protonation, resulting in a steeper slope according to the pH changes (Figure 3g).

### 2.3. Detection of Protein (Albumin) in Urine Sample

Increased urinary protein levels can be a significant indicator of adverse kidney and cardiovascular problems [7,8,80]. Among various proteins, albumin (Alb) is normally found abundantly in the urine; hence, it is standardized as a biomarker for screening proteinuria [81,82]. The Kidney Disease Outcome Quality Initiative of the National Kidney Foundation published clinical practice guidelines for CKD that recommend initial screening and monitoring of urine Alb [83,84,85]. In the early stage of CKD, the amount of Alb in the urine starts to increase due to reduced kidney function, enabling the progression of the disease to be diagnosed by regular measurements of Alb levels. For example, a urine Alb level above 30 mg/g may indicate kidney disease, even if the estimated glomerular filtration rate is above 60. For this reason, various methods to monitor urine Alb have been developed, such as liquid chromatography [86], capillary electrophoresis [87], fluorescent probes [88], and enzyme-linked immunosorbent assay [89]. Although these methods are reliable, they are limited by the complicated sample preparation, expensive pre-treatment, and time-consuming processes. In recent decades, field-effect transistors (FETs) based on single-walled carbon nanotubes (SWCNTs) have been widely used in the growing field of biochemical sensing applications owing to their advantages of easy miniaturization, label-free detection, rapid response, and integrability.

Kim and Kim [35] reported an SWCNT-FET using the bromocresol green (BCG) dye-binding method, which is based on the specific binding between BCG and human serum albumin (HSA) for the detection in urine, as shown in Figure 4a. The authors loaded 0.5 wt% of bovine serum albumin (BSA) as a blocking agent before introducing HSA because non-specific binding can be a significant issue affecting the performance of this type of sensor. As a result, they successfully measured HSA with a LOD of 18.6 μg/L by measuring the electrical conductance of SWCNT-FET, which showed improved sensitivity compared to those of previously developed HSA sensors based on different methods such as total internal reflected resonance light scattering [90], surface-enhanced Raman scattering [91], and zero-current potentiometry [92]. Moreover, the biosensor achieved a real-time response with increasing HSA concentrations from 100 pM to 10 μM in a healthy human urine sample (Figure 4b). Overall, this biosensor exhibited the advantages of improved response time and sensitivity; however, the repeatability and reproducibility for continuous monitoring remain to be investigated.

In 2019, Zhang et al. [36] proposed an electrochemical biosensor using a dual-signal strategy, including a current change of the substrate and solution probe. This biosensor is based on a molecularly imprinted polymer (MIP) for the measurement of HSA in urine using 3D cavities on the surface formed by an HSA template, as shown in Figure 4c. The authors claimed that optimization of various factors and the dual-signal strategy (ΔIsubstrate and ΔIprobe) for detecting HSA in urine offer an opportunity to improve sensing performance; indeed, their biosensor exhibited a wider detection range from 0.1 ng/L to 0.1 mg/L and a lower LOD of 30 pg/L compared with those of previously reported methods for HSA detection. In addition, the biosensor showed acceptable repeatability with a relative standard deviation (RSD) of 4.4%; a lifetime longer than 20 days; and low interference toward other biomolecules such as glycine, glutamate, cysteine, tryptophan, histidine, dopamine, ascorbic acid, hemoglobin (Hb), and BSA.

Karim et al. [37] reported the first FET immunosensor that was based on a high aspect ratio of ZnO nanorods NRs for HAS detection, as shown in Figure 4d. The linearity and selectivity of the immunosensor were enhanced with the addition of the antibody (Ab) that facilitated the immobilization of HSA on the surface of the device. The Ab also acted as an exclusion membrane, simultaneously excluding some interferents such as small molecules, protein fragments (gamma globulin, glycoprotein, ribonuclease, lysozyme, hemoglobin, Tamm–Horsfall mucoprotein, and Bence–Jones protein), and other residues. To detect the presence of albumin, a functionalization process is required on the immunosensor surface. The most commonly used cross-linking process for an immunosensor is the APTES method because its amino group is in one end, and three ethoxy silanes groups covalently attach to surfaces in the other end. The albumin immobilized to the surfaces of the immunosensor to induce a larger signal as an electric double layer (EDL) at the surface–electrolyte interface determines the minimum sensing distance, which leads to a V_TH_ shift in the transfer characteristic according to each binding interaction. Their results demonstrated a high sensitivity of 0.826 mA (g/mL) with a LOD value of 9.81 μg/L in a linear range from 10 μg/L to 100 mg/L (Figure 4e). In addition, their sensor showed good storing stability at 4 °C in the dark for up to 360 days (Figure 6f). Moreover, other proteins in urine such as cytokines, chemokines, and growth factors are also being studied as the biomarker of disease determination [93,94].

### 2.4. Detection of Ketone Bodies in Urine Sample

Urinary ketone bodies are composed of three small compounds, acetoacetic acid (AcAc), 3-β-hydroxybutyric acid (3β-HB), and acetone, which play an important role as indicators of metabolic health status, aiding in the diagnosis of metabolic conditions such as obesity, central obesity, metabolic syndrome, dyslipidemia, and type 2 DM. The normal concentration of ketone bodies in the urine is less than 1 mg/dL. In diabetic ketoacidosis (DKA), due to reduced insulin levels, the increase in ketone bodies in the blood leads to their excretion via the urine, thereby increasing the concentration of ketone bodies in the urine, which is known as ketonuria. Unlike ketone bodies in the blood, 3β-HB is absent and AcAc is relatively abundant in the urine. Therefore, the detection of AcAc in urine could be a clinically useful method to monitor and diagnose DKA. Compared to obtaining blood samples, urine monitoring is a simpler, less expensive, and less invasive method for confirming the presence of AcAc. Despite these advantages, continuous monitoring of AcAc in urine has rarely been studied.

Recently, Go et al. [38] reported a new biosensor with a multi-layer enzyme [NAD+ and a mixture of d-β-hydroxybutyrate dehydrogenase (HBDH) and nicotinamide adenine dinucleotide (NADH)]-modified electrode, which can quantify AcAc in the urine, and is reusable and more accurate compared to the conventional dipstick test (Figure 5a). 3β-HB is transformed from AcAc by catalyzing a reaction between HBDH and NADH, which allows for the transduction of the concentration of AcAc into the current peak at 0.073 V. The sensitivity of the biosensor for detecting the AcAc concentration in phosphate-buffered saline (PBS) solution was experimentally confirmed to be 6.27 mg/dL and a LOD was 6.25 mg/dL in the range of 6.25–100 mg/dL, which was equally able to identify ketone bodies in patients diagnosed with DKA as the urine dipstick test (+++: up to 100 mg/dL; ++: up to 50 mg/dL; +: up to 10 mg/dL; trace 5: 10 mg/mL, and–as the normal level). Moreover, the biosensor was reusable after simple cleaning steps, and a change in the current value of less than 5% was maintained even after 10 uses (Figure 5b). The measurements of urine samples of 20 patients with DKA showed an excellent correlation with the detection of ketone bodies in the same patients using a commercially available dipstick test (Figure 5c). This represented the first attempt to detect ketone bodies in urine using a biosensor in a clinical test.

### 2.5. Detection of Hemoglobin in Urine Sample

One of the most important substances for assessing the physiological condition of the human body is Hb, which is responsible for carrying 97% of the oxygen in the blood; changes in Hb levels in the blood may indicate several diseases [9]. The typical blood Hb level for males and females is 140–180 mg/mL and 120–160 mg/mL, respectively [95]. When the Hb level in the blood is too high (i.e., over the renal threshold), Hb begins to appear in the urine. People who suffer from hematuria show a concentration of ~1.0 mg/mL of Hb in the urine. The presence of Hb in the urine causes a change in the color of the dipstick pad to green/dark blue, which can be associated with kidney stones, renal carcinoma, and other disorders [96]. However, since the dipstick test is based on the presence of peroxidase-like activity and not Hb itself, the presence of myoglobinuria or hypochlorite in urine may cause false-positive results. Thus, a microscopic examination of the urine is required to confirm the presence of Hb and exclude a false-positive result.

To overcome the issue of the false-positive result due to interferential components in urine, Anirudhan and Alexander [39] developed a potentiometric biosensor for the determination of Hb directly in urine using surface-modified multiwalled carbon nanotubes (MWCNTs)–MIP, as shown in Figure 6a. Because of the highly porous and hollow structure of MWCNTs–MIP, it is possible to directly immobilize Hb inside the matrix during sample loading. A potentiometric signal was observed with a linear correlation in the range of 1.0–10.0 mg/L and a LOD of 1.0 mg/L (Figure 6b). In a real urine sample test, the performance of the biosensor was comparable with that of the conventional high-performance liquid chromatography method (97.5% recovery and RSD < 1.0). Furthermore, the selectivity of the MWCNTs–MIP sensor was verified using other proteins such as HSA, myoglobin, and cytochrome C, showing a negligible change compared to Hb (Figure 6c). The authors claimed that since the Hb concentration in urine for people with hematuria is 1 mg/mL, the accuracy of the biosensor with a LOD as low as 1.0 µg/mL is sufficient to determine Hb in urine. Although this study reported promising results for the measurement of Hb, it did not demonstrate the ability for continuous monitoring of Hb concentrations in real urine samples. Han et al. [40] performed another study for Hb determination using a disposable electrochemical sensor, which was based on the reversible redox reaction as shown in Figure 6d. They built an electrode modified with Fc [CO-Glu-Cys-Gly-OH] (Fc-ECG) and Fc [CO-Cys-(Trt)-OMe]_2_ (Fc (Cys)_2_) for the detection of Hb in biological fluids. They confirmed the morphological structure of the electrode by scanning electron microscopy images. This sensor showed a linear response for Hb in the concentration range of 0.1–1000 mg/L and had a LOD of 0.03 mg/L (Figure 6e). To evaluate the selectivity of the sensor towards other biomolecules, differential pulse voltammetry in the presence of potentially interfering molecules such as HSA, glucose, ascorbic acid, IgE, and dl-cysteine was conducted under optimized conditions. Moreover, recovery studies demonstrated that the electrochemical biosensor had high reproducibility with an RSD lower than 2.8% and good recovery of 95.5–103.2% in detecting Hb in human serum. The authors claim that the results of the reproducibility and recovery tests are consistent with the clinical requirements for Hb analysis.

### 2.6. Detection of Nitrite in Urine Sample

UTIs are among the most common bacterial infections seen in the general population. Even though UTIs are rarely fatal, if left untreated, the infection could spread from the kidneys into the bloodstream, causing bacteremia. Although the gold standard for a diagnosis of UTIs is to perform a bacteriological urine culture and physical examination, dipstick tests are widely used in clinical practice owing to their aforementioned advantages of convenience, low cost, and quick response. The parameters of the dipstick test to diagnose UTIs include nitrite, LE, and pH. Among them, nitrite detection relies on the ability of bacteria such as *Escherichia coli*, *Pseudomonas aeruginosa*, *Enterobacter*, *Serratia*, *Citrobacter*, and *Proteus* to convert nitrate to nitrite in the urine; positive results of a dipstick test indicate the significant presence of such bacteria (at least 10^5^ colony-forming units per milliliter) [97]. To date, many studies have been carried out to improve the accuracy of nitrite detection, and the sensitivity and specificity of nitrite detection for UTIs are ~50% (45–60%) and ~95% (85–98%), respectively [98].

Zou et al. [41] reported a graphene electrochemical transistor (GECT) biosensor based on a gold nanoparticles modified reduced graphene oxide (AuNPs/rGO) nanocomposite (Figure 7a). The detection and measurement of their sensor could be employed as gate voltage induced by the electrooxidation of nitrite at gate electrodes. By using this approach, their sensor achieved an excellent LOD of 0.1 nM and exhibited the response over a wide concentration range from 0.1 nM to 7 µM and from 7 to 1000 µM linearly (Figure 7b). In addition, to confirm the high selectivity toward nitrite, they added various interfering species such as K^+^, Li^+^, Ca^2+^, Mg^2+,^ NH_4_^3+^, Cl^−^, NO^3−^, SO_4_^2−^, PO_4_^3−^, CH_3_COO^−^, SO_3_^2−^, I^−^ and glucose in PBS solution. Due to the difference in electrocatalytic reaction mechanisms, the response to the addition of each species except for nitrite was negligible (Figure 7c). Although this study reported promising results for the measurement of nitrite using the GECT biosensor, it did not demonstrate the ability for detecting nitrite in real urine samples.

Cardoso et al. [42] were the first to introduce 3D-printed G/PLA sensors for detecting nitrite in biological samples using multiple-pulse amperometry combined with batch-injection analysis (BIA-MPA). The electrochemical performance of nitrite detection was accomplished by mechanical polishing and solvent immersion of the 3D-printed G/PLA sensor surface [99,100]. The BIA-MPA sensor exhibited high performance in detecting nitrite with a LOD of 0.03 μM (Figure 7d). Moreover, the highly linear behavior of the sensor was shown over a wide concentration range of nitrite from 0.5 to 250 μM (Figure 7e). Estimated recovery values by the standard addition method were between 70% and 90% for nitrite in a urine sample, which is within the allowable range for the analysis of biological fluids [101]. For the clinical test, they measured nitrite levels of approximately 5 mM, which is higher than the level of ~1 mM typically detected in patients diagnosed with UTIs [97]. Overall, this study showed that the 3D-printed-G/PLA sensor exhibited superior analytical characteristics for the detection of nitrite in biological samples; however, more experiments and tests are required for its development for selectivity enhancement.

### 2.7. Detection of Bilirubin in Urine Sample

Bilirubin is mainly formed from the degradation of Hb in the reticuloendothelial system [102], which is transported in the circulation in a form bound to Alb traveling through the blood to the liver. After entering the liver, bilirubin is transported into hepatocytes and conjugates with glucuronic acid. Conjugated bilirubin is also called direct bilirubin, as it directly reacts with diazotized sulfanilic acid [103]. The direct bilirubin concentration in the blood is typically 1–5 μM (0.06–0.3 mg/dL), whereas the bilirubin in urine is not usually detectable [104]. When the liver is damaged, bilirubin can leak out into urine; thus, an excess accumulation of bilirubin in urine, which is called bilirubinuria, can help in the diagnosis or monitoring of problems in the liver. Therefore, screening for bilirubin in urine is useful in detecting liver damage or disease even before other clinical symptoms manifest.

The detection of bilirubin using a dipstick test involves a coupling reaction of a diazonium salt in an acidic medium, producing a diazotization color reaction in the presence of bilirubin. However, these chemical reactions on dipsticks can cause a high rate of false-positive results due to interference of various components in urine, such as indoxyl sulfate, high vitamin C, and nitrite [105]. Accordingly, a positive test result from the dipstick test should be verified with a confirmatory test such as the bilirubin tablet test (Ictotest) [106].

Thangamuthu and coworkers [43] proposed electrochemical sensors for a POC assay to detect bilirubin using MWCNTs and electrochemically reduced graphene oxide (Er-GO) separately deposited on SPEs as shown in Figure 8a. Their sensing system consists of a carbon working electrode with a sensitive surface in which the electrochemical reaction occurs. The sensitivity using MWCNT-SPE and Er-GO-SPE showed a linear range over 0.5–500 µM and 0.1–600 µM with a detection limit of 0.3 ± 0.022 nM and 0.1 ± 0.018 nM, respectively (Figure 8b) Moreover when applied to real human serum samples, the recovery value was found from 94% to 106.5% that represents a very good accuracy.

Rahman et al. [44] were the first to develop an enzyme-free electrochemical biosensor for detecting bilirubin in urine using a glassy carbon electrode (GCE) modified with iron-doped antimony oxide nanorods and Nafion to improve the electrical conductivity and chemical stability owing to the high surface area, as shown in Figure 8c. To confirm the accuracy of the biosensor, they first demonstrated that varying bilirubin concentrations (0.1 nM to 0.01 M) in PBS solution can quantitatively change the I–V response, and an extremely low LOD of 16.5 ± 0.05 pM was achieved compared with those previously reported using other methods (Figure 8d) [107,108,109]. The repeatability, which represents the performance of the biosensor, is the most frequently evaluated parameter of accuracy. The repeatability was investigated by successively detecting 0.1 μM bilirubin in seven experiments, achieving similar current results (RSD = 4.24%, *n* = 7). Finally, the authors demonstrated excellent quantitative (~100%) recovery of the bilirubin concentration from 0.1 nM to 0.1 mM using clinical urine and blood serum samples, indicating that the biosensor does not require re-calibration to improve accuracy. Therefore, the authors concluded that their technique for detecting bilirubin is efficient and reliable.

### 2.8. Detection of Leukocyte Esterase (LE) in Urine Sample

LE is a protein released by white blood cells, thus serving as a biomarker for these immune cells. As an enzyme released by neutrophils, LE is also a widely used conventional biomarker for predicting or differentiating UTIs with the dipstick test. A positive result for LE in a dipstick test indicates an elevated number of leukocytes (a condition known as pyuria), which is often caused by UTIs. Many studies have been carried out to evaluate the accuracy of the dipstick test for detecting UTIs; however, the accuracy remains controversial without an adequate explanation of the cause. Moreover, the dipstick test provides only limited qualitative and semi-quantitative information depending on the concentration. Ho et al. [45] reported a paper-based electrochemical biosensor to detect LE (LE-PAD), which was deposited with mixed 3-(N-tosyl-l-alaninyloxy)-5-phenylpyrrole (PE) and 1-diazo-2-naphthol-4-sulfonic acid (DAS) on an Ag film, exhibiting excellent performance and reliability. This biosensor provides quantitative measurement through increased resistance due to non-conductive azo products produced by LE in the reaction areas of the PE and DAS, as shown in Figure 9a. For analysis of real samples, urine samples collected from 16 patients with suspected UTIs were tested using the biosensor. For a UTIs diagnosis, the resulting biosensor exhibited promising sensitivity and specificity of 87.5% and 92.3%, respectively, with higher accuracy compared with those of 62.5% and 76.9% obtained in the dipstick test (Figure 9b). Moreover, the biosensor did not cause a resistivity change from the interfering molecules in urine, such as uric acid, glucose, urea, or ascorbic acid. Although this study demonstrated an efficient chemiresistive method for improving the detection accuracy of LE in urine with a reliable clinical test, there are still challenges to achieving the continuous, real-time monitoring of LE.

## 3. Detecting Other Biomolecules in Urine Sample

### 3.1. Detection of Cancer Biomarkers in Urine Sample

Cancer is one of the major leading causes of morbidity and mortality of deaths worldwide [110]. The International Agency for Research on Cancer (IARC) forecasted approximately 18.1 million new cases and 9.6 million cancer deaths in 2018 [110]. A variety of biomaterials such as genetic products, endogenous metabolites, peptides/proteins, and nuclear matrix proteins with the potential to serve as cancer biomarkers have been found in urine [111]. Therefore, quick quantizable and non-invasive analysis of urinary cancer biomarkers could play an essential role in early-stage screening and rapid diagnosis [112].

Bladder cancer (BC) and prostate cancer are the most common cancers in the world. The gold standard for clinical diagnosis is a costly invasive procedure that causes uncomfortable such as abdominal pain, and urgent and more frequent urination to patients [113]. To improve a patient’s quality of life and reduce the costs of cancer diagnosis, numerous researchers have been efforts to apply non-invasive diagnostic tests to find promising urinary cancer biomarkers such as cytokeratin-19, telomerase, and especially nuclear matrix protein-22 (NMP-22) [114]

Wu et al. [46] achieved a high sensitivity for NMP-22 using a sandwich-type electrochemical immunosensor, which is a simple, cost-effective, and objective biomarker for detecting BC [115]. Figure 10a illustrates the assembly process of the surface-modified immune sensor. Based on the specific binding force between the antigen (e.g., NMP-22) and antibody (e.g., Ab_1_ and Ab_2_), a sandwiched chemical binding with the structure of Ab_1_/NMP-22/NH_2_-SAPO-34-Pd/Co-Ab_2_ was formed. The detection of NMP-22 could be achieved by catalyzing a reduction reaction of H_2_O_2_ on the electrode surface modified by NH_2_-SAPO-34-Pd/Co-Ab_2_. Their immune sensors verified the detection capability of NMP-22 in a wide linear range of 0.001 ng/mL to 20 ng/mL, resulting in a LOD of 0.33 pg/mL as shown in Figure 10b. Such an improvement in sensitivity was achieved by rGO on the surface of the glassy carbon electrode before covalently immobilizing the antibody (e.g., anti-NMP-22) to the biosensor. Their specific selectivity toward NMP-22 was also confirmed in the presence of potential interference molecules, such as BSA, vitamin C, trioxypurine, and glucose, with low current changes of less than a 5.0% difference for the determination of NMP-22, as shown in Figure 10c. Finally, the authors demonstrated that testing on human urine samples gave a good reproducibility with full recovery in the range from 99.5 to 101.2% (RSD 1.6–6.2%), hence showing promising results for an application for BC detecting immune sensors.

Regarding the sensing mechanism of the biosensor for BC detection, other approaches have been also adopted to increase the sensitivity of the NMP-22 antigen. Recently, Li et al. [47] reported highly sensitive and stable electrochemical biosensors based on IGZO-FETs. For quantitative detection of NMP-22, the IGZO layer was functionalized by NMP-22 antibody through APTES, acting as a cross-linker as shown in Figure 10d. Due to forming negatively charged complexes via chemical bonding between NMP-22 and anti-body on the channel surface, the IGZO-FET biosensors exhibited a current decrease in accordance with increasing of the NMP-22 concentration. From the results, a linear correlation in the range of 0.0001–0.1 pg/mL and an extremely low LOD of 2.7 amg/L were observed. Figure 10e represents the electrical response in real-time at a fixed rate of 0.1 mL/min demonstrating significant current changes related to the different concentrations of NMP-22. Moreover, when applied to real human urine samples, the authors observed much higher NMP-22 from patients than those from the doners, as shown in Figure 10f.

Recently, Yang et al. [48] proposed a biosensor integrated with a machine-learning algorithm based on IGZO-FET to detect BC in clinical urine samples. The proposed biosensor array can be also incorporated with the electrical circuit to achieve a portable device for POC applications. Depending on N-terminal and C-terminal anti-NMP-22 functionalized IGZO-FET, authors demonstrated that the antibody orientation affects conductance changes in the IGZO channel. The linear detection range of anti-NMP-22 functionalized biosensors is from 10^−16^ to 10^−12^ g/mL. Moreover, the proposed biosensor exhibited excellent reproducibility with an RSD of 4.3% in eight experiments, and a good recovery range from 95.3 to 104.3%. Through clinical trials, the authors simultaneously identified five bladder tumor-related proteins, including NMP-22, CD47, CK18, CD47, and CK8. Using the proposed biosensor array, the author demonstrated that the machine-learning algorithm neural network is crucial for selective BC diagnosis. Finally, to enable POC diagnosis, the authors demonstrated integrating the proposed biosensor into an electrical circuit and transmitting the analysis results via a wireless Bluetooth unit.

Pal et al. [49] reported the electrochemical immunosensor-modified gold nanoparticle on the surface of GO to detect prostate-specific antigen (PSA). Figure 11a exhibits the proof of concept of the biosensor to detect PSA. To confirm the stepwise assembly of the Au-GO electrode, they performed electrochemical impedance analysis (EIS) for immobilized molecules as shown in Figure 11b. The EIS analysis exhibited increasing interfacial resistance after immobilized antibodies on the Au-GO electrode surface. The immunosensor showed high performance in detecting PSA with a LOD of 5.4 fg/mL and a linear range from 0.001 fg/mL to 0.02 μg/mL. Moreover, the repeatability was investigated successively in human serum, achieving RSD of 2.91% (*n* = 5) (Figure 11c). Although the electrochemical immunosensor showed good performance in detection of PSA in the clinical sample, there are still challenges to overcome problems such as complicated production processes for immunosensors.

Koo et al. [50] demonstrated an electrochemical sensor based on screen-printed gold electrodes to detect miR-107, a potential biomarker for PSA cancer diagnosis, in human urine samples without any amplification process. To increase adsorption efficiency between miR-107 and gold electrodes, the authors added poly (A) tails to the 3′ ends of miR-107 using poly (A) polymerase enzyme.

The electrochemical detection mechanism of miR-107 adsorbed on the surface of the gold electrode with a [Fe (CN)_6_]^3−/4−^ redox reaction is displayed in Figure 11d. Using this approach, an excellent LOD of 10 fM with a linear range of 5 fM−5 pM was achieved, and these results (Figure 11e) are comparable to previous reports [116,117]. For specific detection of miR-107, as clearly shown in Figure 11f, the results showed miR-107 was about 10-fold higher than other species, such as miR-200c and miR-429. As a result, the authors claimed that the electrochemical sensors could be applied for clinical applications to detect prostate cancer (Figure 11g).

### 3.2. Detection of Inorganic Ions in Urine Sample

Ion concentrations, such as sodium, chloride, and iodine, in urine are also excellent indicators of a patient’s health condition and provide the information required to diagnose illnesses. For example, sodium measurements in urine can help to determine the risk of kidney failure in patients with CKD and the integrity of tubular resorptive function [11]. A certain level of chloride and iodine in the urine can be an indicator of several problems associated with kidney inflammation and increased risk of infant mortality, respectively [15,16,118]. Therefore, to realize urinalysis as a reliable and accurate method for the diagnosis of disease and medical conditions in their early stages, it is important to measure ion concentrations in urine.

Most ion biosensors use Ag/AgCl as reference electrodes owing to their stable potential and low noise level in body fluids. Although various advantages have been proposed in the literature, a major drawback of Ag/AgCl as reference electrodes is that they must be contained in a glass tube and a filling solution is required for precise measurements, which poses a challenge for an integrated sensing system in POC applications. To overcome these difficulties, Oh et al. [51] demonstrated the applicability of a two-dimensional sensing structure using a graphene ion-sensitive FET with an ion-sensitive membrane (G–ISFET-ISM), using fluorinated graphene (FG) as the reference electrode instead of Ag/AgCl for detecting sodium in the urine. As shown in Figure 12a,b, the biosensor consisted of source and drain electrodes of G-ISFET-ISM and a gate electrode of FG; the fluorobenzene of the gate electrode prevented a chemical interaction with other ions by blocking the active group of graphene. Owing to the sodium ion selectivity coefficients of ISM, the biosensor showed good performance in distinguishing sodium ions in urine samples containing other ions such as potassium and calcium. The authors claimed that their biosensor exhibited higher reproducibility compared to that of previously published graphene-based ion sensors using Ag/AgCl electrode [119], gate-free [118], and platinum wire [120] setups, based on the real-time and linear response obtained (Figure 12c). To validate the potential of the biosensor for clinical use, urine samples were collected from four patients to determine sodium ions, which were used without any dilution process. Based on the results of the trial, the authors asserted that the G-ISFET with FG sensor exhibited good sensitivity, linearity, and selectivity owing to its protective effect on other biomaterials such as calcium, hydrogen, potassium ion, and Alb in the urine (Figure 12d). Although the authors demonstrated that the G-ISFET sensor was suitable for identifying sodium ions in urine with good selectivity and sensitivity, they did not show interference results from representative interfering ions such as uric acid, l-ascorbic acid, glutathione, and acetaminophen in urine.

Recently, Lannazzo et al. [52] proposed a combination of electrochemical and optical sensors based on screen-printed commercial electrodes (SPCE) functionalized with graphene quantum dots (GQDs)-15-crown-5 and GQDs-18-crown-6 composites to determine potassium and sodium ions. Compared to the reference sample, the surface-modified GQDs/SPCE was more conductive due to a decrease in the charge transfer resistance confirmed by EIS measurement. The sensitivity for detection of potassium and sodium ions was confirmed to be in the range of 1 to 1000 mM, enabled by chelating ionophores.

Cunha-Silva and Arcos-Martinez [53] proposed a screen-printed platinum electrode (SPPtE) for measuring the chloride content in various samples such as human sweat, plasma, and urine, using interactions between the chloride and platinum electrode surfaces, as shown in Figure 13a. The electrochemically generated chlorine gas in urine, which was released by oxidization of the platinum surface, acts as a partition to prevent the absorption of chloride ions on the platinum surface. Figure 13b exhibits the cathodic stripping voltametric response of the biosensor, demonstrating highly satisfactory analytical performance toward chloride in a linear range from 0 mM to 150 mM and good reproducibility with an RSD of 5.80% (*n* = 7). The SPPtE showed a detection range from 0.76 to 150 mM with a sensitivity of −24.147 µA/mM. Although the authors stated that the SPPtE method is unsuitable for direct analysis in some samples, considering the typical range of chloride concentration in urine (20–40 mM), this method exhibited sufficient sensitivity to measure chloride in the urine.

Khunseeraksal et al. [54] exhibited an electrochemical biosensor that can quantify iodide ions in the urine of pregnant women through a reduction reaction of iodide ions at the GCE surface modified with silver oxide microparticles–poly acrylic acid/poly vinyl alcohol (Ag_2_OMPs-PAA/PVA) as shown in Figure 14a. An advantage of this approach is enhancing detection sensitivity and reproducibility by entrapping nanoparticles and preventing their separation from GCE. This sensor was successfully applied for the detection of iodide ions in the urine of pregnant women in a linear range from 10 to 104 µM and a LOD of 0.3 µM (Figure 14b). In addition, the biosensor showed no interference with iodide ions against the urea, glucose, potassium, chloride, nitrite, sodium, ascorbic acid, and dopamine. However, this biosensor presents limitations with respect to the detection range, real-time response, and continuous monitoring which limit their application prospects.

For real-time, rapid, and quantitative detection of a wide range of iodide ion concentrations, my group developed a low-voltage biosensor based on sol-gel IGZO-EGTFTs [55] (Figure 14c). Unlike the conventional operating mechanism of EGTFT, a negative bias was applied to the Ag/AgCl reference electrode of the biosensor to enhance the electrochemical reaction of iodide ions in the electrolyte. We used IGZO as a channel layer to enhance charge transfers at the interface between IGZO and iodide ions in the electrolyte, and the biosensor can be operated under 0.5 V with a high ON/OFF state current ratio above 10^8^ and a transconductance value of 1.14 mS (Figure 14d). In addition, the interference experiment was performed to estimate the ionic disturbance by other ions such as KCl, NaCl, K_3_PO_4_, L-ascorbic acid, uric acid, acetaminophen, and glutathione. All results indicated a high selectivity toward iodide ions without any ionic disturbance due to the electrochemical reaction process (Figure 14e). For a real-time response, the developed biosensors successfully achieved continuous quantification of iodide ions in artificial urine monitoring based on the World Health Organization guideline (Figure 14f). Therefore, this novel biosensor has the potential to be developed as a tool for the rapid, sensitive, and real-time detection of iodide ions.

## 4. Conclusions and Outlook

In this review, we have described recent technologies developed in the research field of biosensors for the detection of specific biomolecules in human urine, which are typically measured with the dipstick test. Compared with the dipstick test, biosensor technologies are better suited for urinalysis as POC applications with excellent advantages in terms of sensitivity, selectivity, and reusability. Additionally, biosensor technologies for human urine could provide continuous and sustainable health monitoring from the reliable urine sample sources produced by daily user behaviors, unlike sweat, blood, and tissue (skin/hair/flesh/etc.) in practical situations. The developed POC biosensor technologies for clinical applications also could be adopted in near future because of their advantages, such as low cost, ease to use, extreme sensitivity, real-time response, simple sample preparation, and simple instrumentations. However, many challenges remain as POC applications, and various aspects in the field of biosensors still require further exploration.

First, biosensors to detect various low-concentration target molecules in urine, such as human cells, proteins, DNA, miRNA, and ions simultaneously remain challenges. Second, in terms of clinical and/or disease diagnosis, in-depth research regarding interfacial chemical reactions, surface interactions, electronics, and circuits is required based on an amalgamation of underlying studies on biology, chemistry, physics, materials science, nanotechnology, electrical engineering, etc. Because the accuracy of biosensor detection is dependent on the analyte-biosensor interaction, it is also essential to conduct in-depth research on their inherent physical and chemical properties. The precision of biosensors can be improved by enhancing sensing signals toward the target and reducing interference signals. Third, mass-productive and user-friendly biosensors should be developed; a fully automated sample preparation process, and interpretation of various types of diseases. Finally, the development of a self-monitoring system using wearable (implantable) biosensors for clinical application is required for continuous tracking of patient status. The implantable biosensor such as urinary catheters, intraocular pressure, and glucose sensors for diabetics could transmit information including vital value, the concentration of biomarkers, and disease progression in real time. Owing to these advantages, a self-monitoring system could dramatically improve disease management. In the future, we expect that ongoing research and development of biosensor technologies will overcome the current limitations and utilize innovative POC application technologies based on artificial intelligence.

## Figures and Tables

**Figure 1 biosensors-12-01020-f001:**
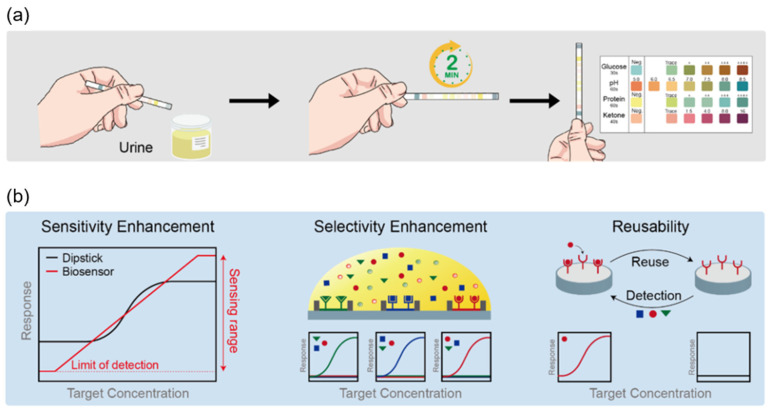
Comparison of point-of-care urinalysis with the dipstick test and biosensor. (**a**) Schematic representation of how to test urine using the dipstick test. (**b**) Schematic representation of the advantages of using a biosensor for urinalysis, enabling improved sensitivity, selectivity, and reusability compared with those of the dipstick test.

**Figure 2 biosensors-12-01020-f002:**
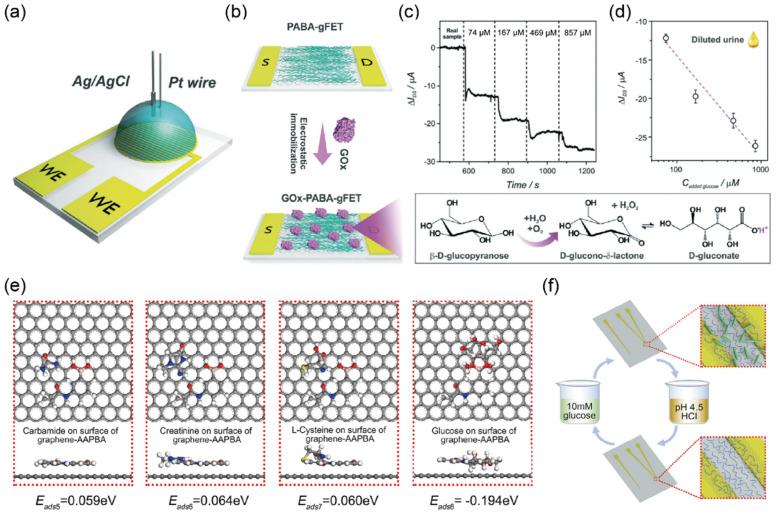
Selective detection of glucose in biological samples using GFET biosensor (**a**) Illustration of the experimental setup to perform poly(3-amino-benzylamine-co-aniline) (PABA) electropolymerization on the graphene field-effect transistor (GFET). Source and drain electrodes are used as working electrodes of a three-electrode cell. The platinum wire and Ag/AgCl are used as counter and reference electrodes, respectively. (**b**) Overview of glucose oxidase (GOx) immobilization in PABA-GFETs and the glucose oxidation reaction. (**c**) Real-time electrical current measurement of the sensor after injection of spiked urine samples (V_G_ = −0.2 V, V_DS_ = 0.1 V) and (**d**) curve fitting using the linear response of sensitivity of −13 ± 2 μA per decade of glucose concentration (R^2^ = 0.9462, *n* = 2) (reprinted with permission from [30]; copyright 2021 RSC). (**e**) The binding energy of 3-acrylamidophenylboronic acid (AAPBA)-functionalized graphene with carbamide, creatinine, l-cysteine, and glucose, respectively. The negative binding energy (−0.194 eV) indicates strong chemical adsorption toward glucose. (**f**) Schematic illustration of the recycling detection of the polymer functionalized GFET with a simple solution process. The covalent bond between glucose and the polymer can be broken under an acid environment. (Reprinted with permission from Ref. [31]; copyright 2022 Elsevier.)

**Figure 3 biosensors-12-01020-f003:**
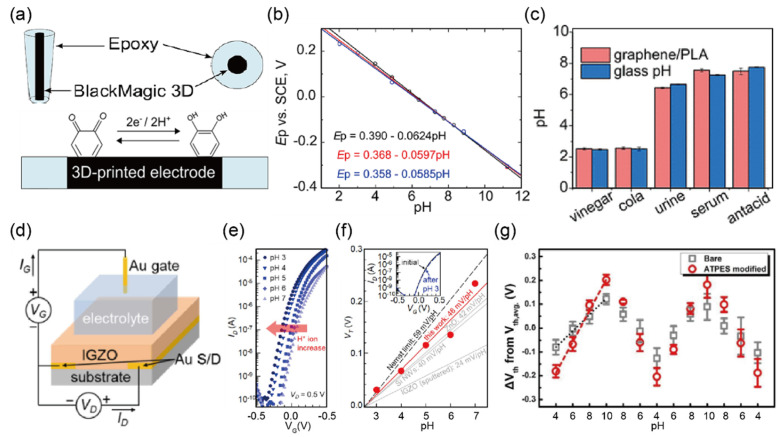
Selective detection of hydrogen ion (**a**) Schematic diagram of the 3D-printed electrode geometry and redox reaction of quinone groups on the electrode surface. (**b**) The relationship between pH and the reduction peak potential of three independent electrodes for the pH range from 2.02 to 11.22 with linearity (R^2^ ≥ 0.99 for all sensors). (**c**) Bar graph comparing the pH detection results obtained by the graphene/polylactic acid (G/PLA) sensor and a conventional glass pH probe in unadulterated real samples, including vinegar, cola, urine, serum, and antacid (reprinted with permission from [32]; copyright 2017 ACS). (**d**) Schematic the indium–gallium–zinc–oxide electrolyte-gated thin-film transistors (IGZO-EGTFTs) structure. (**e**) Transfer characteristics of the IGZO-EGTFTs with varying values of pH. As the pH decreased from 7 to 3, the threshold-voltage (V_TH_) shifted in a negative direction. (**f**) Sensitivity analysis based on the V_TH_ shift of IGZO-EGTFTs, including a slope comparison with the Nernst limit and other technologies. Inset: transfer curves at pH 7 before and after exposure to the pH 3 solution (V_D_ = 0.5 V) (reprinted with permission from [33]; copyright 2021 IEEE Xplore). (**g**) Response of V_TH_ shifts of IGZO-EGFETs with and without amino-silanization exposed to varying pH levels (reprinted with permission from Ref. [34]; copyright 2018 Elsevier).

**Figure 4 biosensors-12-01020-f004:**
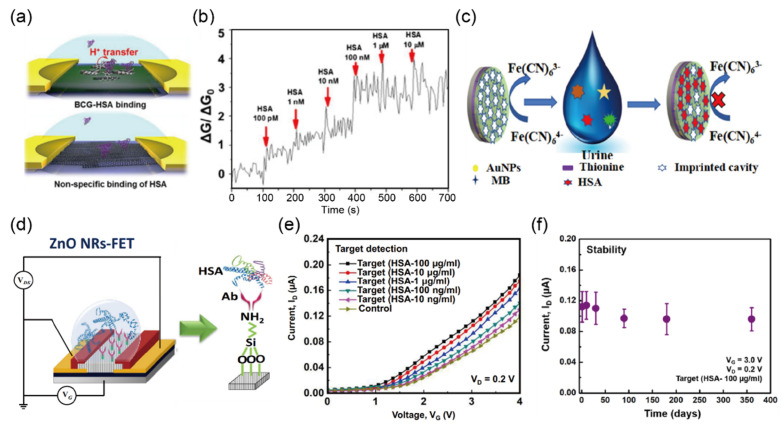
Selective detection of human serum albumin (HSA) in biological samples (**a**) Schematic illustration representing the operational mechanism for bromocresol green (BCG)–HSA binding. (**b**) Measurement of the conductance of the single-walled carbon nanotube (SWCNT)-field emission transistor (FET) in healthy human urine samples (reprinted with permission from Ref. [35]; copyright 2016 Springer). (**c**) Schematic illustration representing the protocol of the molecularly imprinted sensor. (Reprinted with permission from Ref. [36]; copyright 2019 Elsevier.) (**d**) Schematic diagram and electrical connections of the HSA detection mechanism with modified Ab/APTES. (**e**) Transfer characteristics with different HSA concentrations from 10 ng/mL to 100 µg/mL (V_D_ = 0.2 V). (**f**) Storing stability test of the ZnO NRs-FET sensors for up to 360 days for the detection of 100 µg/mL HAS (V_G_ = 0.5 and V_D_ = 0.2 V) (reprinted with permission from Ref. [37]; copyright 2021 Springer).

**Figure 5 biosensors-12-01020-f005:**
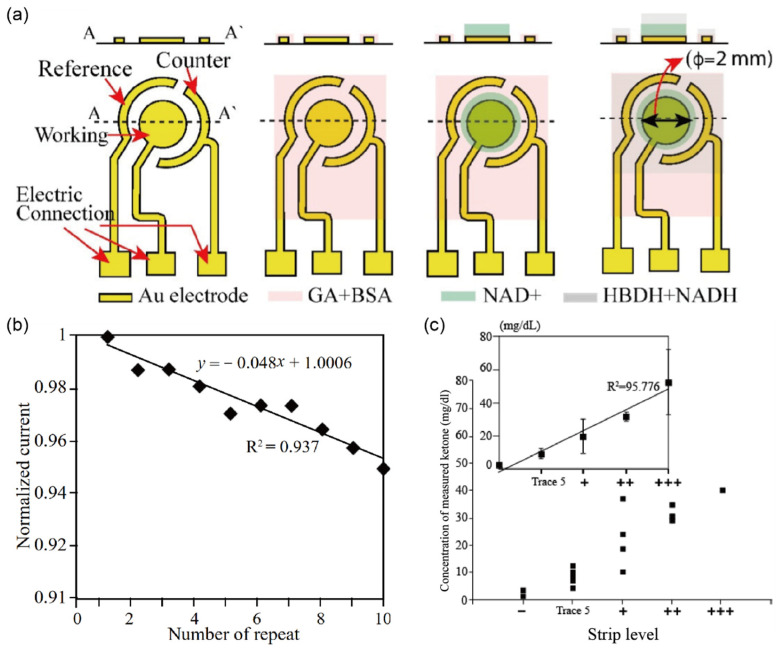
Selective detection of ketone bodies in clinical samples using multi-layer enzyme biosensors (**a**) Schematic diagram and layout of the electrode (working, counter, reference) structure. A Cr/Au layer with 50/100 nm thickness was fabricated by the conventional semiconducting process. (**b**) Comparable CV measurement during 10 repeated tests using 50 mg/dL of AcAc solution. The degradation of signal intensity is less than 5%. (**c**) Measured current plot comparing the electrochemical sensor and dipstick acquired from ketone levels (+++: up to 100 mg/dL; ++: up to 50 mg/dL; +: up to 10 mg/dL; trace 5: 10 mg/mL, and – is normal level) in 20 patients, (reprinted with permission from Ref. [38]; copyright 2021, MDPI).

**Figure 6 biosensors-12-01020-f006:**
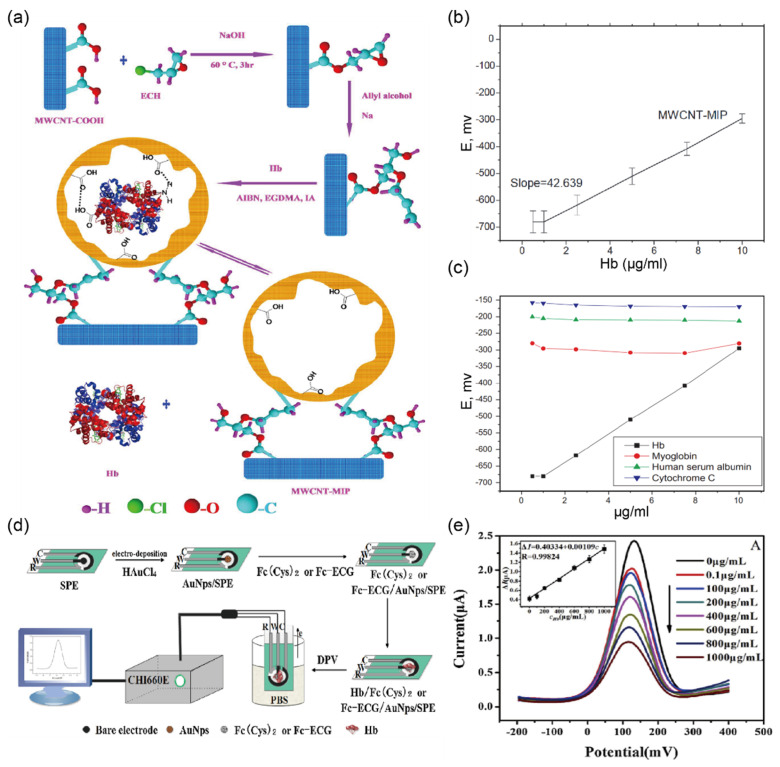
Selective detection of hemoglobin (Hb) in clinical samples (**a**) Proposed protocol of multi-walled carbon nanotube (MWCNT)-molecularly imprinted polymer (MIP) sensors for Hb detection. (**b**) The calibration curves reflect the response of the MWCNT-MIP sensors. (**c**) Evaluation of the selectivity of the MWCNT-MIP sensor toward myoglobin, human serum albumin, cytochrome C, and Hb. The potentiometric response showed no significant effects compared to Hb detection using the MWCNT-MIP sensor (reprinted with permission from Ref. [39]; copyright 2017, Elsevier). (**d**) Illustration representing fabrication and test environment setup for Hb detection by differential pulse voltammetry (DPV). (**e**) DPV response of the Fc-ECG/AuNPs/SPE electrochemical sensor against the concentration of Hb in the range of 0 to 1000 µg/mL. The inset shows a linear calibration curve for different concentrations of Hb. (Reprinted with permission from Ref. [40]; copyright 2019, Elsevier.)

**Figure 7 biosensors-12-01020-f007:**
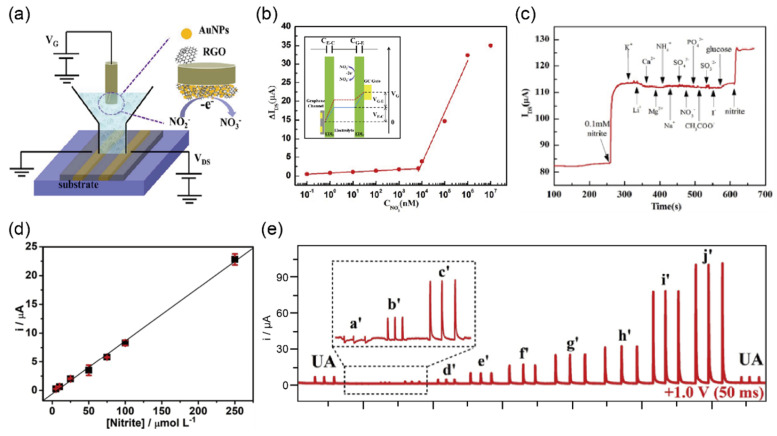
Selective detection of nitrite in biological samples (**a**) Illustration representing the structure and electrical connections of the graphene electrochemical transistor (GECT) sensor. (**b**) Calibration curve of changes of channel current (ΔI_DS_) versus concentration of nitrite from 0.1 nM to 7 µM (V_D_ = 0.2 V). Top-left inset shows the sensing principle of GECT toward nitrite. (**c**) Interference test with various ions such as K^+^, Li^+^, Ca^2+^, Mg^2+,^ NH_4_^3+^, Cl^−^, NO^3−^, SO_4_^2−^, PO_4_^3−^, CH_3_COO^−^, SO_3_^2−^, I^−^, and glucose against 0.1 mM nitrite (reprinted with permission from Ref. [41]; copyright 2019, Elsevier). (**d**) Calibration curve of different concentrations of nitrite from 0.5 to 250 μM. The limit of detection (LOD) is 0.03 μM with linearity (R^2^ ≥ 0.9986). (**e**) Amperometric measurement operated at an applied potential of 1.0 V (50 ms each) with triplicate injections of nitrite in uric acid solution (a′–j′: 0.5–250 μM) (reprinted with permission from Ref. [42]; copyright 2020, Elsevier).

**Figure 8 biosensors-12-01020-f008:**
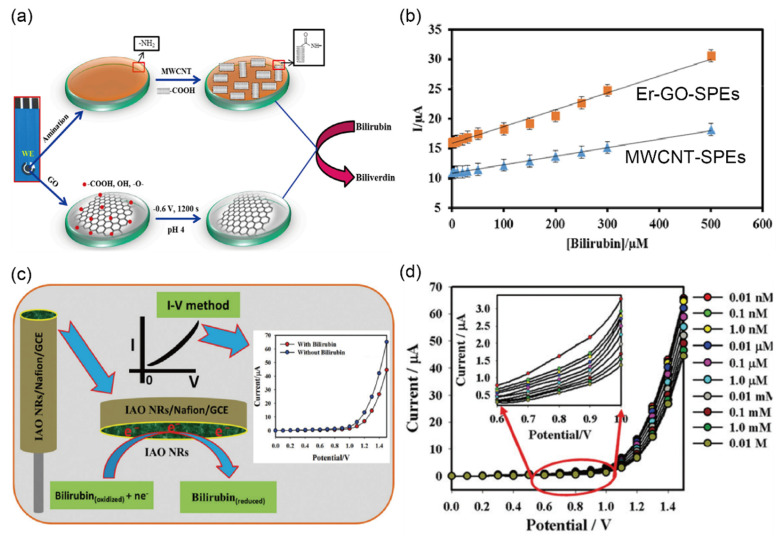
Selective detection of bilirubin (BR) in clinical samples using electrochemical biosensors (**a**) Schematic diagram representing a proposed procedure for multi-walled carbon nanotubes (MWCNTs) screen-printed carbon electrodes (SPEs) and electrochemically reduced graphene oxide (Er-GO) SPE bilirubin sensors for detecting BR. (**b**) Calibration curve of MWCNT-SPEs and Er-GO-SPEs showing the current response at various concentrations of BR (reprinted with permission from Ref. [43]; copyright 2018, MDPI). (**c**) Illustration of the sensing flow to detect BR on a modified glassy carbon electrode (GCE). Top-right inset shows the I–V response with (red dots; 1 µM, 25 µL) and without (blue dots) BR from aqueous solutions. A significantly low current response was observed due to the reduction reaction of BR on the IAO NRs/GCE surface. (**d**) I–V response curve of different aqueous BR concentrations from 0.01 nM to 0.01 M in 5.0 mL of PBS with various potentials from 0.0 to 1.4 V at −0.2 V steps. The current response slowly decreased as the concentrations of BR increased. The inset shows the magnified view of the current variations for +0.6 to +1.0 V (reprinted with permission from Ref. [44]; copyright 2019, RSC).

**Figure 9 biosensors-12-01020-f009:**
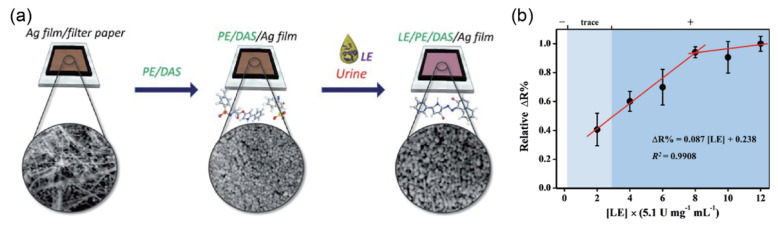
Selective detection of leukocyte esterase (LE) in clinical samples using electrochemical biosensors (**a**) Schematic diagram illustrating the experimental design with a chemiresistive method for the quantitative detection of LE in urine. (**b**) Relative resistivity calibration curve of LE-PAD toward different LE concentrations. The limit of detection was 1.91 (×5.1 U mg^−1^mL^−1^; 20 WBC per mL). All resistivity responses are presented as mean values ± standard deviation (SD), *n* = 3, and the corresponding points are overlaid. The correlation of the resistivity response was 99% (reprinted with permission from Ref. [45]; copyright 2020, RSC).

**Figure 10 biosensors-12-01020-f010:**
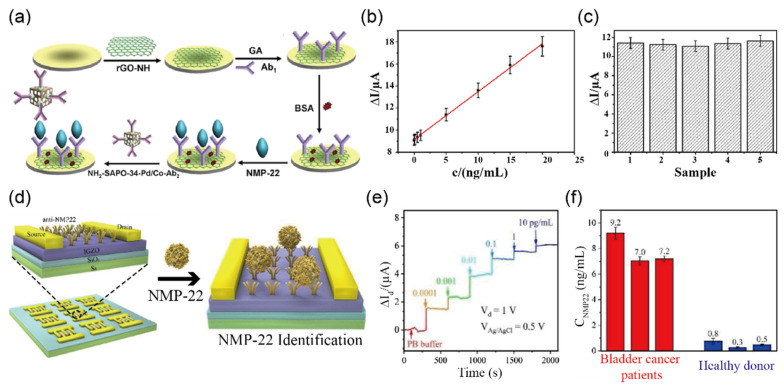
Selective detection of biomarkers for diagnosis of bladder cancer in clinical samples (**a**) Illustration representing the electrochemical mechanism for detecting NMP-22 relying on NH_2_-SAPO-34-Pd/Co-Ab_2_ (**b**) Linear responses of the immunosensor with NMP-22 concentrations ranging from 0.001 to 20 ng/mL; LOD and correlation coefficients are 0.33 pg/mL and 0.998, respectively. (**c**) Evaluation of the selectivity of the immunosensor toward NMP-22 (5 ng/mL) (1) solutions containing 500 ng/mL of interfering substances such as bovine serum albumin (2), vitamin C (3), trioxypurine (4), and glucose (5). Error bar = RSD (*n* = 5). (Reprinted with permission from [46]; copyright 2016 NATURE.) (**d**) For bladder cancer detection, schematic diagram of the IGZO-FET biosensor array functionalized by anti-NMP-22 (**e**) Real-time response of IGZO-FET toward different concentrations of NMP-22 from 0.0001 to 10 pg/mL. The drain current (I_D_) was measured at V_DS_ = 1 V and V_GS_ = 0.5 V (**f**) Evaluation of sensing performance toward different concentrations of NMP-22 in response to clinical urine samples from bladder cancer patients and healthy donors (reprinted with permission from Ref. [47]; copyright 2020, Elsevier).

**Figure 11 biosensors-12-01020-f011:**
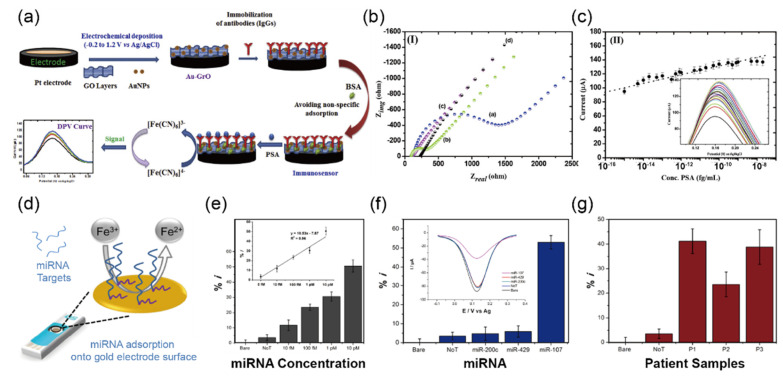
Selective detection of biomarkers for diagnosis of prostate cancer in clinical samples (**a**) Schematic illustration of the fabrication and experimental process for electrochemical sensor based on Au-graphene oxide (GO) electrode for detecting prostate-specific antigen (PSA). (**b**) For electrochemical impedance spectroscopy analysis, Nyquist plots of (**a**) Pt, (**b**) Pt-Au (**c**) Pt-Au-GO, and (**d**) Pt-Au-GO-monoclonal anti-PSA antibody showing the real (Ζ′) and imaginary (Ζ′′) parts of impedance for the Pt-Au-Go-monoclonal anti-PSA antibody in PBS (100 mM, pH 7.4). (**c**) Calibration plots showing current change toward various concentrations of PSA from 0.001 fg/mL to 0.02 μg/mL. LOD for PSA was found to be 5.4 fg/mL (inset: the higher magnification of the DPV curve). (Reprinted with permission from Ref. [49]; copyright 2020, Elsevier.) (**d**) Schematic illustration of the operational mechanism of the electrochemical sensor to detect target miRNA by absorption on the gold electrode surface. (**e**) Corresponding DPV signal change on the concentration of miR-107 in the range from 10 fM to 10 pM. (**f**) Evaluation of the selectivity towards miR-200c, miR-429, and miR-107. The DPV response indicated no significant effects compared to miR-107 detection. The error bars represented the RSD of three independent experiments (inset: the DPV response). (**g**) Measured miR-107 level in three prostate cancer patients using the electrochemical sensor. (Reprinted with permission from Ref. [50]; copyright 2016, ACS.)

**Figure 12 biosensors-12-01020-f012:**
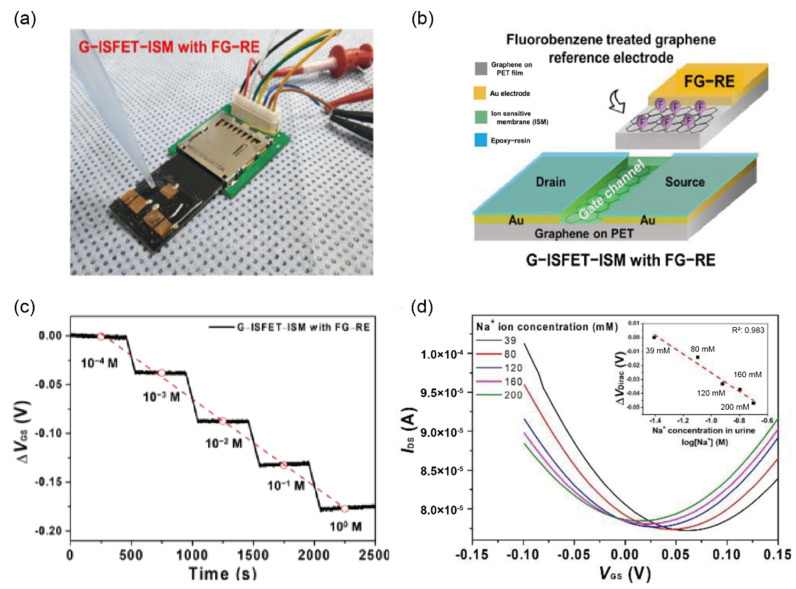
Selective detection of sodium ions in clinical samples (**a**) Schematic diagram of the experimental setup and (**b**) biosensor components of a two-dimensional structure composed of G-ISFET-ISM and FG-RE for detecting sodium ions on the PCB designed as an SD card. (**c**) Real-time response of G-ISFET with FG-RE toward different concentrations of sodium ions from 0.1 mM to 1 M. The gate voltage (V_GS_) was measured at V_DS_ = 0.05 V and I_DS_ = 180 µA. (**d**) Corresponding transfer characteristics are dependent on the concentration of sodium ions in real human patient samples. The top-right corner inset shows the linear relation between the shift at the Dirac point (V_Dirac_) and the concentration of sodium ions with a sensitivity of −0.29 mV/mM. (Reprinted with permission from Ref. [51]; copyright 2021, MDPI.)

**Figure 13 biosensors-12-01020-f013:**
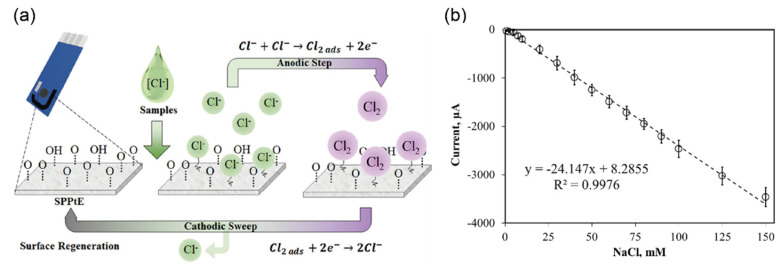
Selective detection of chloride ions in clinical samples (**a**) Schematic diagram showing the electrochemical mechanism for detecting chloride ions relying on the chloride complexation at the electropolished screen-printed platinum electrode (SPPtE) surface. (**b**) Linear responses of the SPPtE sensor with chloride ion concentrations ranging from 0 to 150 mM, registered for seven distinct SPPtEs; the relative standard deviation (RSD) and sensitivity are 5.8% (*n* = 7) and −24.147 µA/mM, respectively. (Reprinted with permission from Ref. [53]; copyright 2019, Elsevier.)

**Figure 14 biosensors-12-01020-f014:**
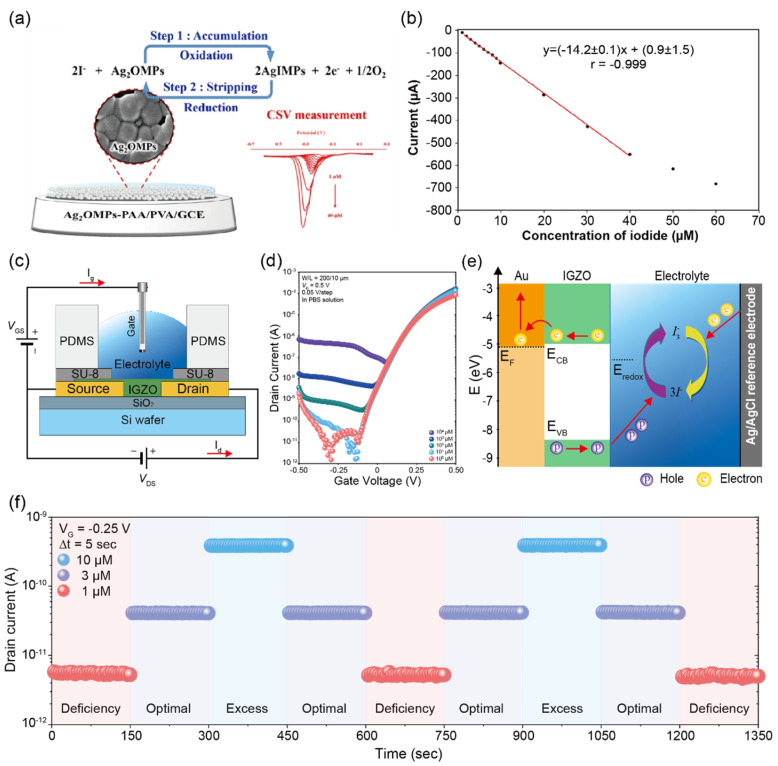
Selective detection of iodide ions in clinical samples (**a**) Schematic diagram of detection principle for iodide ions sensors. (**b**) Changes in the current of the biosensor with increasing iodide concentration under optimal conditions. Standard calibration curves of iodide were plotted with different iodide ion concentrations in the range from 1 to 40 µM (reprinted with permission from ref. [54]; copyright 2020, Springer). (**c**) Schematic diagram of the device structure and electrical connections. Ag/AgCl reference and Au electrodes are used as the gate and source/drain electrodes, respectively. (**d**) Transfer curve of IGZO-EGTFT with different iodide ion concentrations in the range from 10 to 10^4^ µM in PBS solution. The drain current was measured at V_G_ = −0.3 V and V_D_ = 0.5 V. (**e**) Energy band diagram of Au/IGZO-electrolyte-Ag-AgCl showing the hole and electron transport path by the redox reaction of iodide ions. (**f**) Real-time source-drain current response dynamic switching every 5 s for the iodide ions status in samples from school children based on the World Health Organization guideline. (Reprinted with permission from ref. [55]; copyright 2022, Elsevier.)

**Table 1 biosensors-12-01020-t001:** Summary of biosensors developed to identify urine-based biomarkers that are typically detected by dipstick tests.

Biomarker	Dipstick	Typical Range	Biosensors					
			Sensing Material	Detection Method	Linear Range	LOD	Human Sample	Ref.
Glucose (mM)	Negative, 5, 15, 30, 60, 110	0–0.8 mM	GOx–PABA-GFET ^c^	I–V	10 μM–1 mM	4.1 μM	urine	[30]
			P-GFET	I–V	0.04–10 mM	1.9 μM	urine	[31]
pH	5.0, 6.0, 6.5, 7.0, 7.5, 8.0, 9.0	4.5–8	3D-printed G/PLA ^d^	CV ^n^	2.02–11.22	-	urine	[32]
			IGZO-EGTFT ^e^	I–V	3–7	-	-	[33]
				I–V	4–10	-	-	[34]
Protein (g/L)	0.15, 0.3, 1.0, 3.0, 20	Less than 30 mg/g	BCG-modified ^f^ SWCNT- FETs	Conductance	0.07–70 mg/L	18.6 μg/L	urine	[35]
			GE/AuNPs/PTH-MB/MIP ^g^	DPV ^o^	0.1 ng/L– 0.1 mg/L	30 pg/L	urine	[36]
			ZnO NRs-FET	I–V	10 μg/L– 100 mg/L	9.81 μg/L	serum	[37]
Ketone (mg/dL)	5, 15, 40, 100	Less than 1 mg/dL	Enzyme-depositedAu electrode	CV ^n^	6.25–100 mg/dL	6.25 mg/dL	urine	[38]
(Cell/μL)	Non-hemolyzed: 10, 80, 100 Hemolyzed: 25, 80, 200	0–5 Cell/μL	MWCNT-MIP	Potential measurement	1.0–10 mg/L	1.0 mg/L	urine	[39]
			Fc [CO-Glu-Cys-Gly-OH]	CV ^n^	0.1–1000 mg/L	0.03 mg/L	serum	[40]
Nitrite	Negative, Positive	Less than 7 μM	Au/rGO-GECT ^h^	I–V	0.1 nM–7 μM 7–1000 μM	0.1 nM	-	[41]
			3D-printed G/PLA	Amperometry	0.5–250 μM	0.03 μM	urine	[42]
Bilirubin (μM)	17, 35, 70	-	Er-GO	I–V	0.1–600 μM	0.1 ± 0.018 nM	urine	[43]
			IAO-NRs/Nafion/GCE ^i^	I–V	0.1 nM–0.01 M	16.5 ± 0.05 pM	urine	[44]
Leukocyte esterase (WBC/μL)	25, 75, 500	-	PE/DAS/Ag/film ^j^	Resistivity	20–80 WBC/μL	20 WBC/μL	urine	[45]
NMP ^a^-22		Less than 5 ng/mL	Less than 5 ng/mL	NH_2_-SAPO-34-Pd/Co-Ab_2_	CV ^n^	0.33 pg/mL	urine	[46]
			IGZO-FET	IGZO-FET	I–V	2.7 amol/L	urine	[47]
			IGZO-FET	IGZO-FET	I–V	3.2 × 10^−17^ g/mL	urine	[48]
PSA ^b^		4–10 ng/mL	4–10 ng/mL	Au-GrO	DPV ^o^	0.24 fg/mL	serum	[49]
miR-107		-	-	Screen-Printed Gold Electrodes	DPV ^o^	10 fM	urine	[50]
Sodium ions		40–220 mmol/day	40–220 mmol/day	G-ISFET ^k^ with FG-RE	I–V	-	urine	[51]
Potassium ions		0–10 mM	0–10 mM	Graphene Quantum Dots	Potentiometric	1 mM	-	[52]
Chloride ions		20–40 mM	20–40 mM	SPPtE ^l^	CV ^n^	0.76 mM	urine	[53]
Iodide ions		0.3–6.0 μM	0.3–6.0 μM	Ag_2_OMPs-PAA/PVA ^m^	CSV ^p^	0.3 μM	urine	[54]
			IGZO	IGZO	I–V	1 μM	-	[55]

^a^ NMP: nuclear matrix protein; ^b^ PSA: prostate-specific antigen; ^c^ Gox-PABA-GFET: glucose oxidase-3-amino-benzylamine-co-aniline-graphene field-effect transistor; ^d^ PLA: graphene/polylactic acid; ^e^ EGTFT: electrolyte-gated thin-film transistor; ^f^ SWCNT: single-walled carbon nanotubes; ^g^ GE/AuNPs/PTH-MB/MIP: gold electrode/Au nanoparticles/polythionine-methylene blue/molecularly imprinted polymers; ^h^ GECT: graphene electrochemical transistor; ^i^ GCE: glassy carbon electrode; ^j^ PE/DAS: 3-(N-tosyl-l-alaninyloxy)-5-phenylpyrrole/1-diazo-2-naphthol-4-sulfonic acid; ^k^ ISFET: ion-sensitive field-effect transistor; ^l^ SPPtE: screen-printed platinum electrode; ^m^ Ag_2_OMPs-LAA/PVA: silver oxide microparticles-poly acrylic acid/poly vinyl alcohol; ^n^ CV: cyclic voltammetry; ^o^ DPV: differential pulse voltammetry; ^p^ CSV: cathodic stripping voltammetry.

## Data Availability

Not applicable.

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
