# Peer review of "Recent Advances in Biosensor Technologies for Point-of-Care Urinalysis"

_biosensors, 2022, doi:10.3390/bios12111020_

Round 1
Reviewer 1 Report
The manuscript titled "Recent advances in biosensor technologies for point-of-care urinalysis: Systematic review" by Hwang et al. focuses on the biosensor technologies for urinalysis. Authors described the use of different biosensing devices for the measure of analytes, commonly conducted using the dipstick test. I suggest to accept after minor revision.
Minor comments:
Table 1 is not so clear, I suggest to improve adding orizontal lines, also I suggest to describe in a legend at the bottom of the table, the abbreviations of the sensing materials (such as GFET: Graphene Field Effect Transistor) and detection methods (such as DPV: Differential pulse voltammetry), change "Real Sample Analysis" in "Human Sample" and remove human from the column, leaving just urine or serum.
Figure 2. It is not clear if figure 2 is created with the data of 635 or 20 articles. Could authors define this point? Because if including the 635 articles (most of them duplicates, unrelated and on others biofluids), the figure 2 is not representative of the research trends in publications associated with biosensors in the field of urinary biomarkers.
Reviewer 2 Report
This manuscript presents a comprehensive review of various point-of-care (POC) biosensing technologies that could be used in urine biomarker detection. The authors started with a discussion of the limitations of the conventional dipstick test in urinalysis, then reviewed the biosensors based on different target analyte types, and finally concluded with their perspectives on future challenges and development towards POC urinalysis
Overall, the manuscript presents good logic and systematic methodology of the literature survey with clear figures and tables. However, this review paper does not discuss any uniqueness of urine biomarker detection compared to other body fluids like blood, saliva, sweat, etc. Furthermore, the review missed several important analytes in urine such as various protein biomarkers other than albumin, DNA/RNAs, and failed to capture important diseases like early cancer screening or diagnosis. The mere focus on the biosensor’s sensitivity, selectivity, and reusability largely masked other key factors like the sensor’s usability, equipment maintenance, and price in real clinical settings which all play a critical role apart from the sensor’s performance. Therefore, I would recommend this paper for major revision. Several detailed questions/comments in the following should be further addressed:
Comments:
1. What would be some unique clinical challenges of urine biomarker detection compared to other body fluids like blood (serum and plasma), saliva, and sweat? The authors did not provide enough discussion in this unique biofluid, for example, biomarkers type and concentration, patient's hydration level, sample collection, and storage, etc
2. The protein detection in urine only covers albumin and hemoglobin and missed other important biomarkers like cytokines, chemokines, and growth factors.
For example
a. Prasad, S., Tyagi, A. K., & Aggarwal, B. B. (2016). Detection of inflammatory biomarkers in saliva and urine: potential in diagnosis, prevention, and treatment for chronic diseases. Experimental Biology and Medicine, 241(8), 783-799.
b. Liu, B. C., Zhang, L., Lv, L. L., Wang, Y. L., Liu, D. G., & Zhang, X. L. (2006). Application of antibody array technology in the analysis of urinary cytokine profiles in patients with chronic kidney disease. American journal of nephrology, 26(5), 483-490.
3. The review failed to cover the DNA or RNA detection in urine, as well as capture important diseases like early cancer screening or diagnosis.
For example
a. Truong, M., Yang, B., & Jarrard, D. F. (2013). Toward the detection of prostate cancer in urine: a critical analysis. The Journal of urology, 189(2), 422-429.
b. Goodison, S., Rosser, C. J., & Urquidi, V. (2013). Bladder cancer detection and monitoring: assessment of urine-and blood-based marker tests. Molecular diagnosis & therapy, 17(2), 71-84.
4. Could the authors evaluate the POC biosensing technologies in terms of other important factors in real clinical settings like the sensor’s usability, whether it requires a well-trained technician to operate it, equipment maintenance, sample preparation, and potential contamination, price, etc, other than just the sensitivity, selectivity, and reusability?
5. The reusability drawback of the dipstick versus other POC biosensors is an unfair claim due to the significant low cost of the dipstick (no need to be reused) and other considerations (e.g. sample contamination, infectious diseases). Suggest removing sentences like “However, the reusability of the dipstick test is limited owing to the single-use test pads and reagents.”
6. Frankly speaking, I don’t think “Biosensor technologies are limited to distinguishing different biomolecules simultaneously; thus, at present, only a single target biomolecule can be detected.” are challenges for the urine biosensors as there are so many protein microarrays or multiplex biosensing techniques published in public or get on the market. (For example, the reference that I mentioned in Comment 2b)
Please think deeper and wider, focusing more on clinically practical challenges. Any chance to develop a continuous urine biomarker monitoring system through a urine catheter that patients frequently use in-hosipital?
Reviewer 3 Report
Dear authors,
please find attached comments and recommendations.
Best regards

Reviewer 4 Report
Hwang et al. presents a systematic review of the biosensor technologies available to identify urine-based biomarkers that are typically detected by the dipstick test. Manuscript is very well written; however, few points need to considered
1. Please check thoroughly for grammatical error.
2. Define how this manuscript overcome the limitations of already presented review paper.
3. Add section describing limitations and future prospects.
4. Include 2022 papers also in references
Reviewer 5 Report
This article does not seem to be for a systematic review about urinalysis, but it is more like a comparative review with dipstick test. The topic itself is very valuable, however the organization of the article was still need to be improved. Here gives some suggestions for further revision:
1. Introduction section should be revised according to a systematic review about urinalysis.
2. The review does not adequately summarize the most commonly utilized techniques to detect urinary biomarkers. Urinalysis includes physical analysis of urine by observing color, odor, specific gravity, etc. Chemical analysis is performed in urine by examining urine pH, protein concentration, blood traces, etc. Microscopic analysis is also performed for screening the existence of bacteria, casts, crystals, cells, etc.
3. The physiological ranges of typical urine biomarkers should be added.
4. “POC urinalysis methods” section should be added. Various POC urinalysis platforms such as dipstick, LFA, paper-based, and microfluidic assays, and analytical devices should be added and discussed.
5. A comprehensive outlook on associated challenges in POC urinalysis of biomarkers should be provided.
Round 2
Reviewer 2 Report
Thank you for the revised manuscript. It looks good to me now. With the new section on urinary cancer biomarker detection and diagnosis, the impact of this review will be strengthened.
Reviewer 5 Report
The present manuscript have gained significant improvement after revision. I recommend publication to the Biosensors.